# Can We Understand Non-Tourism as a Form of Sustainable Tourism? The Role of Lifestyle and Motivations behind Non-Traveling Based on the Hungarian Example

**Lóránt Dénes Dávid [1], János Csapó [1,]\* , Ákos Nagy [2] and Mária Törőcsik [2]**

[1]  Department of Tourism, Constantine the Philosopher University in Nitra, SK-949 74 Nitra, Slovakia; ldavid@ukf.sk

[2]  Institute of Marketing and Tourism, University of Pécs, HU-7622 Pécs, Hungary; nagya@ktk.pte.hu (Á.N.); torocsik.maria@ktk.pte.hu (M.T.)

\*  Correspondence: jcsapo@ukf.sk

**Abstract:** This paper aims to study non-travelers in order to try to understand why they are absent from tourism and what the causes are for their decision. Our research showed that the study of postmodern causes apart from classic ones holds unique potential in the research of sustainable tourism processes as well. The results of cross-tabulation and correspondence analysis show that postmodern and classic causes are tightly connected to lifestyle, which represents the central theme of the current study based on the results of a Hungarian representative online survey. A certain limitation is that our research is based on the case of Hungary; however, the introduced methodology can be used in general for identifying and evaluating non-travelers. As research implications, the authors believe that the methodology and results can be used by the actors of the tourism supply market and by decision makers as well, especially for segmenting purposes. If we understand who the non-tourists or non-travelers are, we can, on the one hand, determine the latent tourism potential of a tourism destination; on the other hand, we can also receive information on specific market segments, which could contribute to sustainable tourism mostly because of the postmodern causes for non-traveling.

**Keywords:** non-tourism; lifestyle groups; travel habits; Hungary; representative study

## 1. Introduction

Market research is primarily interested in the consumers of a particular market and analyzing their consuming behavior and decision-making criteria. Publications that focus on non-purchase/non-consumption have recently been gaining ground in opposition to the consumer society [1–3] or in support of non-development deriving from non-consumption [4–7]. From time to time, advocates also make a stand arguing that getting to know non-purchasers has as much potential as is gained by more traditional analysis of the market, such as by coaxing consumers of the competition or by encouraging more intensive consumption [8]. In their work introducing the concept of the blue ocean, Kim and Mauborgne pointed out the innovative opportunities of non-purchasers [8,9].

In classic marketing literature, non-purchase is discussed partly as specific cases of demand [10] and partly when investigating situations that require, for social (counter marketing) and corporate (demarketing) reasons, steps to cut back demand [11]. The most detailed documentation is concerned with reducing the demand for goods and services deemed socially harmful, especially by encouraging non-purchase (smoking, drugs, obesity) and by exploring marketing effects [12–14].

As far as tourism is concerned, non-purchasers are identified as consumers who are currently potential purchasers in a particular market, are known to the market players, but who still fail to become consumers, as well as those who currently consume in different markets and are not yet present in the minds of market players as potential entities in tourism [15]. This concept represents the notions of market decision makers, rather than an explanation embedded in the motivations of consumers.

Owing to current international and Hungarian trends in tourism [16], the change in the behavior of the consumer, that is, the tourist, and its new tendencies exert an ever-increasing effect in the formation and transformation of tourism [17]. Yet it needs to be pointed out, based on the literature survey of the authors, that the international literature of the field and studies in general are concerned primarily with those who travel and much less, if at all, with the causes of non-travel. It is our conviction that the phenomenon can be a fascinating aspect of non-tourism, arising as it does as a side effect of extensive tourism.

Based on this view, the current paper undertakes to present the characteristics that appear in tourism among non-purchasers as well as the ways of defining and thereby classifying the motivational background of these groups. In addition, we seek to discuss virtual tourism as well, a non-purchase and sustainable option that potential consumers may engage with in novel ways. Acknowledging the practical and theoretical currency of this theme, we aim to present the phenomenon and trends of non-tourism, focusing on classical and postmodern skippers and their lifestyle groups, addressing practical and theoretical questions alike. The introduced methodology can be used in general for identifying and evaluating non-travelers and so provide a better understanding about the different segments of latent and/or potential tourists. We especially aim at understanding as well as proving the relationship between lifestyle groups and non-tourism attitude.

## 2. The Theoretical Framework of Non-Tourism and Lifestyle Correspondences

### 2.1. The Theoretical Framework of Non-Tourism and Non-Travel

When surveying the international literature of this domain, we have found, except for the general theory of tourism [18–21], a lack of studies about non-tourism or non-travel analyzing the topic from a general, motivation-based perspective [22,23]; nevertheless, there are a limited number of works related to the analysis of micro-perspectives on destination level [24–26]. Researchers generally tend to prefer motivation-driven segmentation of current supplies [27,28], and deal to a much lesser degree with latent supply [29]. Naturally, the classic, supply-based division of markets is just as valid for tourism: existing markets, potential markets, and latent markets [15]. Our paper deals with the second and third of these, as existing markets are present, in one way or another, in the processes of tourism.

Within non-tourism, a non-tourist/non-traveler in a potential market can be motivated for travel: although the individual in question can be assumed to possess some of the three prerequisites of tourism (motivation, discretionary income, and free time), he or she does not participate in such processes [30]. It is these causes that are the focus of our current study and future research as well. As regards latent markets, the potential contact with tourism is not yet perceived, a state that, however, can change at any time, giving way to the "blue ocean"—for example, in the extreme case of space travel [31,32].

To be able to explore the more complex causes of non-travel, one must be informed about the characteristics of travelers, the most crucial being, apart from physiological and sociocultural traits, existential attributes [33–35]. As from our point of view the essential ones are those that help explain non-consumption, the emphasis is laid on exploring inhibiting mechanisms (including attitudes, health conditions, special mental states, family responsibilities, and steps limiting one's ability to leave or enter a country).

According to the above, the authors classified the forms of non-tourists and non-travel from the point of view of consumer motivation. Based on this, we can distinguish between classic skippers (non-travel groups that are already known from previously studies) and postmodern ones, whose

absence from classic tourism where one seeks out to change one's environs is caused by entirely different reasons (see Figure 1). The resistance of this group has recently come into focus.

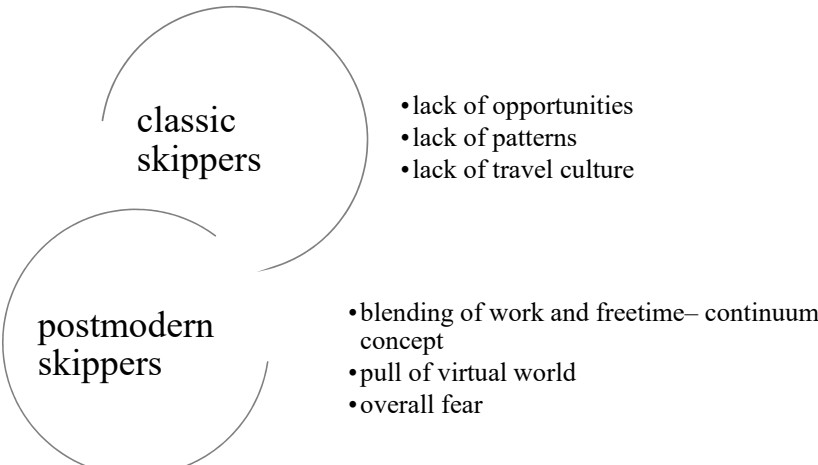

**Figure 1.** The groups of non-tourists, Source: Authors.

Classic skippers are known from several studies, mainly publications that include a statistical apparatus, which often merely state what percentage of a target group does not purchase/consume. This group is hardly ever brought under further investigation except when making stereotypical statements about them. What characterizes classic skippers more than anything else is a combination of lack of finances, scarcity, and an attendant special decision-making pattern [36]. Quite simply, they cannot afford such expenses, although causes can be any type of lack, including that of socialization: they do not have a travel culture and relevant patterns [37]. Often, even vacation time at home is combined with work, in the form of an activity for making money or renovating the home, sometimes even working on tasks of a major undertaking. In addition to the scarcity of discretionary income, classic causes of non-travel include age, health status, and regional–sociological aspects, meaning that limited spending willingness and classic non-travel appear as the combined result of a peripheral socio-economic status and regional–sociological conditions [38]. It can also be triggered by lack of knowledge and lack of companionship.

When postmodern skippers decide not to purchase, it is caused by lifestyle and a conviction about consumption, rather than by limits of economic and financial status. No doubt they, too, are limited in some ways, but with them, we need to take into account the effects of disrupted social frames, disappearing market boundaries, and events that call into question one's security: their staying at home is motivated by reasons entirely different from those in the classic group [39]. For example, more and more studies explore the notion that today's opportunities befuddle those that try to separate work and free time, as they are in constant flux, with one or the other in the foreground at any one time, making classic holidays all but impossible. That there are people who prefer virtual, rather than actual, tourism is primarily due to the technological revolution, especially the spread and high-quality projection and enjoyment of virtual reality [40,41]. Also at play are factors such as an increasing fear of danger and the cumbersome side effects of tourism, making those planning to travel have second thoughts.

As highlighted in the title of the paper, it is an interesting and up-to-date topic whether we can connect non-tourism and non-travel with the aspects of sustainability. It can be stated that the related extensive international literature mainly discusses the topic from the point of view of the already realized tourism processes, focusing as a starting point on the negative impacts of tourism and/or new initiatives and approach to responsible forms of tourism both on the supply and the demand side [42,43], but lacks the analysis of how non-tourism and non-traveling can be considered as a sustainable way of tourism. From the very few examples, Li et al. (2015, 2016) understands non-tourists as a huge potential market that accounts for a large proportion of the population and studies this

area with the aim of converting non-tourists into actual tourism consumers [22,23]. Inspired by this approach, the authors of this publication intend to turn back this idea, since another obvious reason for the relationship between non-tourism and non-travel is that the potential tourists will remain potential and stay at home instead of burdening the environment with their travels. As mentioned above, the motivations for non-traveling can be connected to the classic motivational factors [44–47] of lack of money, lack of time, and lack of travel motivation, but the postmodern reasons provide very interesting aspects for the research of this topic as well. These reasons can be connected to safety issues [48–50] and/or the pull of the virtual world as well [41,51,52].

### 2.2. The Constructs of Lifestyle Research

Lifestyle is a social trait based on a basic human need that represents the concurrent desire to integrate (to belong somewhere) and to differentiate (to be unique). It shows cultural and subcultural behavior patterns within a community, describing the everyday life and preferences of members of the group [53–55]. Although lifestyle research, similarly to the stratification studies of sociology, describes the structure of society, its main concern is everyday esthetics, the values of group members, and consumption decisions [56,57]. Based on their individual decisions, it presents a comprehensive view of consumers' lives and values. The reason for the relative stability of lifestyle groups and constructs is that these groups are formed alongside value choices, often by taking into account another dimension as well. That is why in lifestyle studies a longer period of time can elapse before the need for replication, as we can hardly assume that people keep changing their values every year [58].

Social structure has mainly been a concern for sociologists [59,60], although its importance was quickly realized by marketing experts as well: this wealth of knowledge presents in a well-shaped manner the value preferences inherent in the decisions of groups, a factor in all lifestyle models. It appears that today's marketing professionals are less concerned with lifestyle than they were a decade ago [61], even though a new aspect often surfaces, such as the sameness of real and virtual lifestyles [62] or its different and unique traits [63]. Nevertheless, what can be witnessed today is a general disillusionment, with less support for the validity of lifestyle groups adequately indicating social structure [64].

Another aspect of the issue is that conducting a lifestyle study requires too many resources in terms of both the data collection and analysis phases, and results do not seem to be specific and useful enough in business decisions [65]. With the appearance of scenes [66], groups with unique life views, many argue that classic lifestyle research methods cannot describe society [67]. Attention is moving away, as a result of big data, toward personas [68] and toward the identification of types derived from large databases in order to describe their behavior in certain fields. Despite these criticisms, suggestions, and phenomena, lifestyle is a stable construct that can be applied to consumers along both vertical and horizontal social dimensions [65,69].

Apart from the generally applied dimensions of modern and traditional value orientation, lifestyle research recently has adopted others, representing the specialty of an approach: social status, individuality, and so forth. The present paper deals with the concept of the most recent LifestyleInspiration model [70]. According to the basic tenet of the model applied, we need to incorporate tempo of life (which can be fast or slow) owing to the changes in consumer behavior at the turn of the millennium. Those in the fast lane make more money, possess more knowledge, and can use modern technology (ICT) easily. Those in the slow lane either choose this to show being anti-mainstream or are passed by the dominant trends of society. Thus, the theoretical underpinning of the model states that, apart from value orientation, lifestyle is determined by tempo of life [71].

As the starting point, groups are placed in the four fields of a two-by-two matrix: fast—modern (moving with the trends), slow—modern (preferring anti trends), fast—traditional (active members of the social majority), and slow—traditional (skippers). The first studies reporting on the 2003 LifestyleInspiration model research that used face-to-face survey and focus group respondent data from 4000 participants appeared in 2004 [72]. The applicability of the model was verified [73], followed

in 2013 by the study of the lifestyle of Generation Z [74]. One of the results of this study involving the young generation is that social media, internet use, mobile phones, and a growing narcissism were novel characteristics that today are far from relevant only to the youth.

The authors of the paper applied the latest LifestyleInspiration study, because it dealt with the target country of the present paper [55,75–77]. From the countrywide sample of 2000 participants between the ages of 16–74—who were reached by using the method of personal inquiry—the answers of those respondents were subjected to analysis, where the applied attitude assertions turned out to be relevant after the consistency study, and thus the size of the sample came down to 1832. The authors used the range of human values developed by Schwartz [78] to determine the values of value orientation, and revalidated attitude assertions were adopted to define life tempo values. Factors were taking shape to create the two dimensions by making a main component analysis on the basis of 40 attitude assertions (14 life tempos and 16 values). The analysis revealed the following individual dimensions:

- traditional value system: honesty, environmental protection, pursuit of happiness and freedom, security, giving assistance, and respect for traditions
- modern value system: change, adventure, creativity, success, and acquiring wealth and power
- fast life tempo: ambitious way of thinking, digitalization, seeking the new, anxiety, and an active lifestyle
- slow life tempo: passive and slow lifestyle, absence, striving for permanence, and adherence

The aim of the dimensions defined by factor analysis was to identify typical lifestyle groups according to the base model. According to the chosen viewpoints, these groups are internally homogeneous but externally can be clearly separated from each other. A K-means cluster analysis was conducted to carry out the analysis in order to identify nine groups. The results are shown in Figure 2.

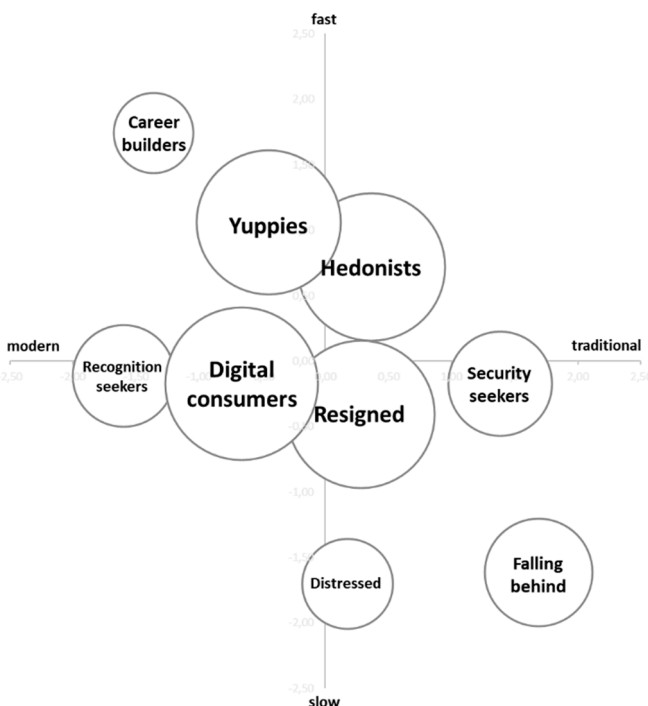

**Figure 2.** Lifestyle groups in Hungary in 2018 (n = 1833), Source: Authors.

It can be seen from the position of the groups that they are concentrated along three levels. The fastest and probably (then demonstrably) wealthiest are the career builders, the hedonists, and the yuppies. The social mean is represented by recognition seekers, digital consumers, the resigned, and security seekers. The distressed and those falling behind are in the most problematic situation.

The features of these lifestyle groups are listed below, focusing on general group traits [75].

*Career builders* (5.2%). This group is characterized by a particularly modern and fast life tempo, also explained by the demography of the segment, most of them young, under the age of 34, who are either doing their studies and/or working. The members of this cluster make their purchase decisions in a favorable financial situation.

*Yuppies* (16.7%): They also live at a fast life tempo and espouse modern values, and the majority of them are in an economically active life phase, although they might also be doing academic studies. In this group, there are middle generation members living in a family context, whose ratio is higher than among career builders.

*Digital consumers* (18.8%): They clearly espouse modern values but represent a lifestyle at an average life tempo. The members of the group, mainly under 34 years of age, are characteristically unmarried, and a lower percentage of them are married. Although young, they either do active physical work due to their low education levels or are unemployed. As a result, their income is unstable, limiting their consumption and efforts despite their desires, and they are bound to tightly restrain their purchases. As they put it, they are closely related to modern technologies, which they arguably do not use as a tool for their work, but rather for entertainment.

*Recognition seekers* (8.5%): They prefer modern values, and their life tempo is not fast enough to satisfy their aspirations as a group of the unsatisfied. A group of mainly middle-aged males (above 35 years of age), they are characterized by aspiring for better living standards and the recognition of their environment, yet are unable to achieve these goals, because they lack persistence and the ability to properly exploit opportunities. A high number of them are divorced.

*Hedonists* (17.5%): A group mainly including older members who espouse traditional values. The members of the cluster generally come from the middle generation, between the ages of 35–45, who mostly have families and usually do not live alone. For the most part, graduated women with active intellectual occupations who can save money alongside a good level of daily expenses fall into this category.

*Resigned* (17.6%): They show an average life tempo, having an inkling of the traditional and slow category with slightly depressive members. Including a high rate of divorced or widowed women above 45, group members usually have an intermediate level of education and average income, but due to their health conditions or age many are economically inactive or retired.

*Security seekers* (8.8%): Holding traditional values, they live at an average life tempo, and thus a slow life conduct is not characteristic of them, despite their mature age. Most members are women over 45, married, or perhaps divorced or widowed. Retired members also appear among them, but they can make ends meet from their income provided by their intermediate or high educational level. A part of the group can also save money.

*Distressed* (6.7%): This category can be characterized as the slowest living in towns, and their circumstances are precarious from every aspect. A group that seems deflated, and can be considered neither traditional nor modern, although—due to their age and circumstances—they regularly live at a slow life tempo. Its members are mostly above 55, living alone, and their children (if they have them) do not live with them anymore. They tend to be low-skilled and many are retired, which is the reason we find people with poor financial conditions in this category.

*Falling behind* (9.4%): The category of mainly the old generation living in the countryside who have slow life tempo and practice traditional values, a characteristically retired group whose members live alone (widowed) and are mainly above 65 years of age, living under extremely limited financial conditions, similarly to the distressed.

## 3. Materials and Methods

After providing an overview of the relevant literature and advancing the theoretical framework of the theme, the paper investigates the relationship between non-tourism, non-travel, and the previously

introduced lifestyle groups. In order to identify these groups and the aimed relationship, a quantitative analysis is needed, which can be further assessed later on through qualitative methods.

*3.1. Sample Selection*

We hired a research agency, which conducted a face-to-face survey that was representative for the Hungarian population aged 16 to 74 in terms of gender, age group, and location of residence (Appendix A) (stratified sampling). (It was also agreed to exclude sampling bias.) Hungary was chosen as a case study for practical reasons, but as a less developed country, its society consists of people who follow various lifestyles where modern and traditional values also appear. Although the authors carried out the survey in this country, we strongly believe that the presented methodology can be used in other regional contexts as well, especially in Central or Eastern Europe. The survey in April and May 2018, employing a sample of 2001 people, aimed to collect data on respondents' habits and opinions concerning a range of topics including ageing, health care innovations, tourism, eating habits, cultural activities, and lifestyles. We have followed the principles of the Singapore Statement on Research Integrity. The paper focuses on the tourism component of the results.

*3.2. Statistical Procedures*

Results have been analyzed and will be presented in two stages. First, we will introduce the primary (frequency) data, followed by demographic background analyses using cross-tabulation, especially Chi-squared tests. Statistical methods have been applied to investigate whether there are statistically verifiable, significant differences among the answers given by respondents in the various demographical groups. Variables included gender, generation, type of residence location, level of education, and the subjective judgement of one's financial status. This stage is followed by interpreting the results of the correspondence analysis conducted on lifestyle groups and causes of non-travel. Correspondence analysis is a useful tool to uncover the relationships among categorical variables; it is a multivariate graphical technique designed to explore relationships, and this is exactly what we have aimed at in our research—to explore the relationship between lifestyle groups and causes of non-travel.

## 4. Results

*4.1. Presentation of the Empirical Results in Connection with Non-Traveling*

We need to emphasize as a starting point of the analysis that 43.3% of the Hungarian respondents have gone on a holiday or rest by travel (for multiple days) in the past year (preceding the year of the analysis), with the remaining 56.7% not engaged in any sort of tourism travel.

Further, 30.5% of the classic skippers, the non-travelers, mentioned lack of money and 20.3% lack of time, which were revealed as the main reasons for classic absence together with those who, as they claimed, do not like to travel (10.3%), with lack of company (8.3%) also referred to as a significant cause. Although visible only to a lesser degree, mention must be made of those who do not travel due to cumbersome traffic (6.7%) and those who addressed their health conditions (4.4%) as the reason. While only at a seemingly negligible rate (2.8%), the reasons for postmodern absence also emerged as a cause in people who are afraid of traveling to foreign places (2.9%) and those who reported that they can get travel experiences while staying at home with the help of the internet and technology. Therefore, virtual tourism and the role of emerging digitalization have already come up in the responses (see Figure 3).

Despite the fact that the topic was a novelty for respondents, who had only marginal knowledge with a high rejection rate, more than 10% of them would prefer choosing virtual tourism because of the possible risks of traveling (with scores of 4 and 5 on the Likert scale). Respondents who reported a much higher rejection rate (with 78% rating 1 and 2) tended to disagree with the idea that a virtual travel experience would be more exciting than a real one. However, it might be an interesting result for

researchers of tourism and technology development or the future that respondents had a high interest in "travel" experiences provided by virtual reality, almost one-fifth of them (19.8%).

In the following, respondents' answers according to differences in their financial, educational, generational, and residential circumstances will be explored.

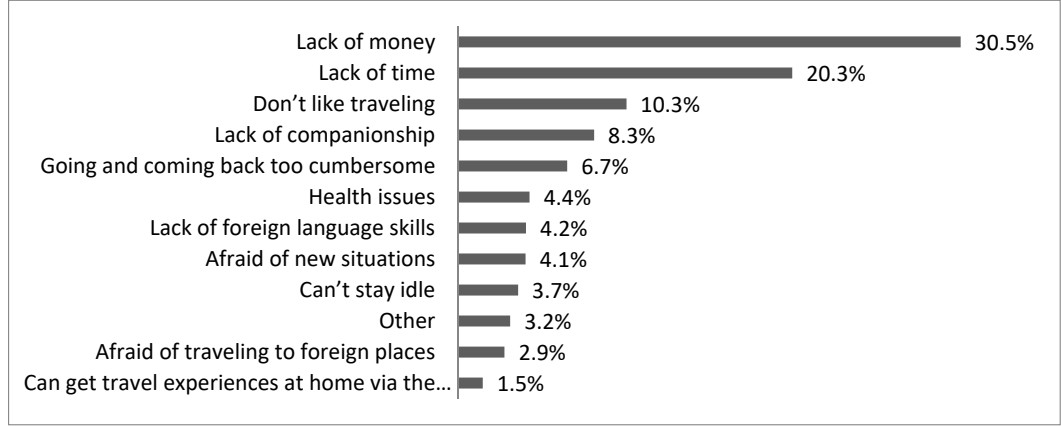

**Figure 3.** The causes of non-travel (What are the reasons why you do not travel? Please choose the three most important reasons from the list below). Source: authors' research, 2018 (with 2328 total choices—each respondent could choose a maximum of three reasons).

As was expected, in terms of the financial circumstances of travelers and skippers, mainly those people travel whose financial situation is grounds for them to do well or very well and who can also save money. However, it should also be emphasized that the least well-off segments also participated in touristic activities. By contrast, the connection between financial conditions and non-traveling can easily be pointed out in terms of non-travelers ($\chi2 = 177.47$; df = 5; $p < 0.000$; Cramer's V = 0.31), revealing the situation of classic skipping.

The analysis of reasons for non-travel and level of education shows that those with a higher education level refer to the reasons of absence the least, as they are the ones who travel the most in the classification of educational qualification ($\chi2 = 169.19$; df = 7; $p < 0.000$; Cramer's V = 0.304). It is characteristic for the ones who have elementary education to mention different reasons for non-travel at the highest rate. Certainly, this result also mirrors the lower financial conditions of this segment, as, among those who have elementary education, there is a significantly higher rate of people whose income is yet to cover the costs of living.

The classic causes of non-tourism in the context of the type of residence were also explored ($\chi2 = 25.29$; df = 4; $p < 0.000$; Cramer's V = 0.117). From this aspect, the main reasons of non-travelers who had a rural, countryside background were lack of time and money, which can be explained either by the time-consuming feature of rural life, or by the less-favorable financial conditions of country folk. In terms of the subjective judgement of their income, country people are significantly different from townsfolk: those who can save money after monthly expenditures are below the average rate. Additionally, people whose income is just enough to make ends meet are above the average rate among the ones living in villages. Incidentally, for people in the capital, these factors were the least frequently cited as reasons for non-travel.

When analyzing the notion of non-travel as a generational dimension, one can find rather straightforward reasons for the sharply different results (the groups are the young generation up to age 29, the middle generation between 30–59, and the elderly, above 60). Non-travel caused by lack of time is mainly relevant for the young, who are followed by the middle generation, and this reason for non-travel appears less frequently among the elderly. In contrast, health problems, naturally, mostly prevent senior citizens from traveling, and the young the least. The same goes for the elderly generation being more likely to be bothered by possibly cumbersome travels or lack of comfort.

Again, staying at home also due to technology appears in terms of this group of questions. The young generation, although at a low rate (4.8%), indicated that "today I can access travel experiences via the internet and technology" as a reason to stay at home. The same answer was selected by 2.8% of seniors, and 3.4% of the middle generation. As we will see later on, the young generation is much more interested in the experience provided by virtual reality. One may assume that the value represented here is associated with the security needs of the older generation ("I can check any kind of destination safely at home by using the net"). The spread of internet use can also play a role.

*4.2. The Relationship between Non-Travelers and Lifestyle Groups*

It seems that there is clear evidence that the rate of travelers is higher in the groups living a faster life and enjoying better financial circumstances. Therefore, the chance of travel has the highest potential in the case of the career builders, yuppies, and hedonists, whereas the groups of those falling behind, the distressed, and recognition seekers tend to be most absent from the travel experience (see Figure 4). The case of recognition seekers is worth particular emphasis, because they are younger than the other two groups close to them (those falling behind and the distressed), and it seems likely that it is mainly their financial status that produces a restrictive effect (this idea will be verified later on).

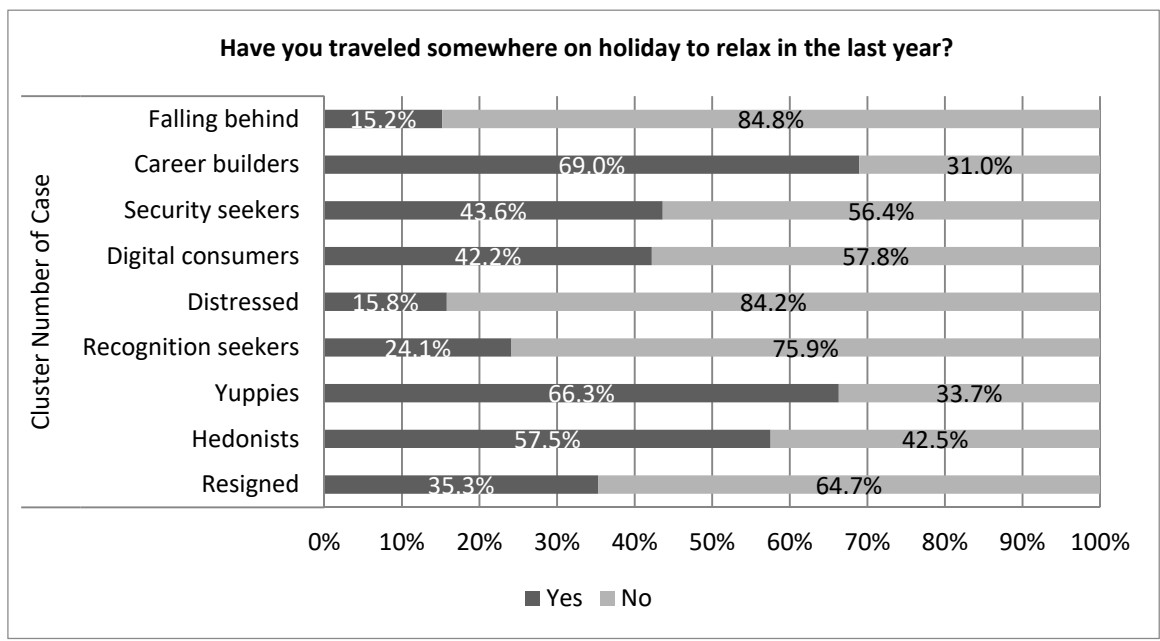

**Figure 4.** Lifestyle groups and travel/non-travel, n = 1833, Source: authors' research, 2018 (N = 2328 (each respondent could choose a maximum of three reasons)).

In order to see the specific reasons for non-travel, we can rely on the data presented in Table 1, which shows what the members of the individual lifestyle groups identified as the primary reasons for not traveling. Basically, it is the lack of time that non-traveling members of all groups pointed out, which might be true but can just as well be a favorable explanation covering the real reason.

As for career builders, apart from lack of time (67.9%), financial constraints also appear, if only at a rate of 18% of the skippers. It is interesting to point out that only the members of this group acknowledged lack of foreign language skills as a factor (3.6%). For them, lack of companionship also raises a problem (3.6%, and this comes up as an issue among security seekers, too). A total of 66.7% of yuppies indicated lack of time, and 22.9% referred to lack of finances.

**Table 1.** Non-tourists' most important absence reasons by lifestyle group, n = 1832. Source: authors' research, 2018.

| | Resigned | Hedonists | Yuppies | Recognition Seekers | Distressed | Digital Consumers | Security Seekers | Career Builders | Falling Behind | Total |
|---|---|---|---|---|---|---|---|---|---|---|
| No choice made | 0.5% | 0.8% | 1.0% | 1.9% | 1.0% | 2.2% | 2.4% | | 3.0% | 1.5% |
| Lack of time | 48.9% | 66.4% | 66.7% | 40.2% | 24.7% | 51.6% | 40.0% | 67.9% | 17.0% | 45.6% |
| Don't like traveling | 14.4% | 9.8% | 3.1% | 9.3% | 24.7% | 14.3% | 22,4% | 3.6% | 33.3% | 16.1% |
| Lack of money | 27.7% | 17.2% | 22.9% | 44.9% | 32.0% | 30.8% | 25.9% | 17.9% | 36.3% | 29.4% |
| Lack of companionship | 2.7% | 0.8% | 2.1% | | 2.1% | 0.5% | 4.7% | 3.6% | 2.2% | 1.8% |
| Going and coming back too cumbersome | 0.5% | 1.6% | 2.1% | | 4.1% | 0.5% | 2.4% | 3.6% | 0.7% | 1.3% |
| Afraid of new situations | | | | | | | 1.2% | | | 0.1% |
| *Can't stay idle* | | | | 0.9% | | | 1.2% | | 0.7% | 0.3% |
| Afraid of traveling to foreign places | 1.1% | 0.8% | | | 1.0% | | | | 3.0% | 0.8% |
| Lack of foreign language skills | | | | | | | | 3.6% | | 0.1% |
| Health issues | 1.1% | | | 0.9% | 10.3% | | | | 3.0% | 1.6% |
| *Can get travel experiences at home* via *the internet and technology* | | | 1.0% | | | | | | | 0.1% |
| Other | 3.2% | 2.5% | 1.0% | 1.9% | | | | | 0.7% | 1.3% |

Attraction to virtual tourism also appeared amongst them ("I can collect travel experiences with the help of the internet and technology"), although only in 1% of claims. Recognition seekers were the group with the highest (44.9%) rate of mentioning of lack of money. Those falling behind (33.3%), the distressed (24.7%), and security seekers (22.4%) said at a very high rate that they do not even like traveling. Since two of these groups are considered to be disadvantaged, their claim seems rather as a means of excuse, which can partially be characteristic of security seekers, too (with 1.2% of this sole group reporting fear of new situations). Of the distressed, 10.3% referred to their health conditions as an impediment to travel, whereas 3% of those falling behind referred to being markedly afraid of traveling.

We could have analyzed the reasons mentioned as second and third as well; however, these are not going to be explored in detail within the present study. Yet we should note that yuppies also join the supporters of virtual tourism—even if only 1% of them. When virtual tourism is analyzed as the third reason of absence, we can find relatively high rates: it is mentioned by 10% of hedonists and 9% each of yuppies and of recognition seekers, with 7% of security seekers and career builders choosing this particular option as a reason for being absent from traveling.

A further analysis of the data was conducted to reveal non-traveling people's attitudes toward travel itself. The results showed a relationship between choosing various reasons (lack of money, time, language skills, and companionship) and agreeing mostly or completely that traveling and holiday-making makes life fuller (60%), even though traveling and holiday-making do not tend to be part of their lives (20–25%). This makes it clear that classic skippers choose not to travel not because they would not want to or because they do not see its importance, but because their life conditions result in their non-travel.

To be able to explore and clarify the relationships between the various lifestyle groups and the causes of non-travel, a correspondence analysis was performed [79]. A correspondence analysis provides a means of graphically representing the structure of cross-tabulations so as to shed light on the underlying mechanisms (see Figure 5). In a typical correspondence analysis, a cross-tabulation table of frequencies is first standardized, so that the relative frequencies across all cells sum to 1.0. One way to state the goal of a typical analysis is to represent the entries in the table of relative frequencies in terms of the distances between individual rows and/or columns in a low-dimensional space [80].

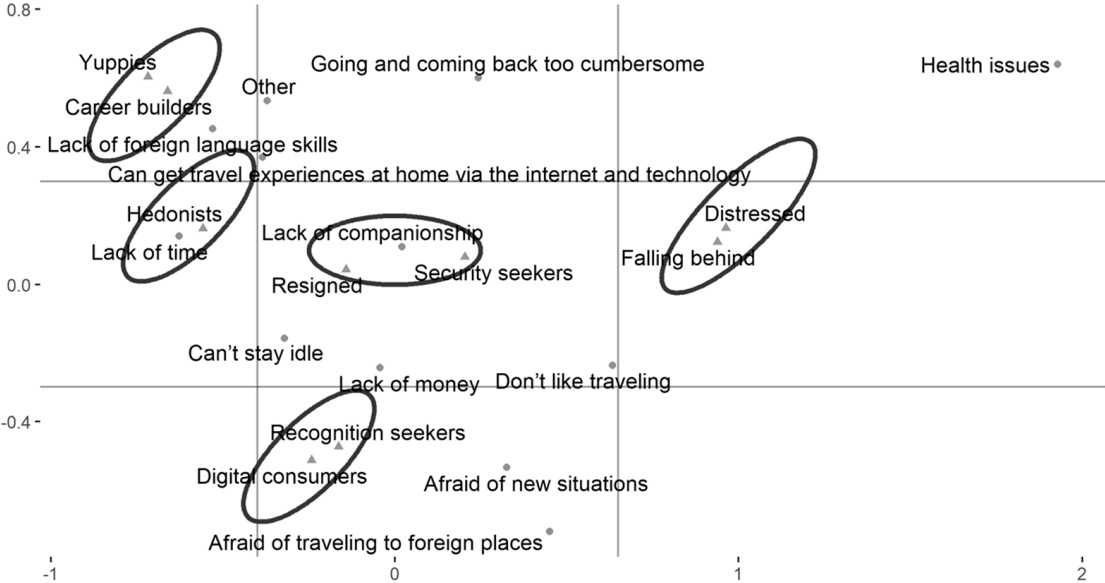

**Figure 5.** The result (perceptual map) of the correspondence analysis. Source: authors.

When interpreting the correspondence map, we can clearly see groups that more closely belong to non-travelers, while moving together with causes also creates larger islands. As for groups with the most travels—career builders, yuppies, and hedonists—the causes of non-travel tend to be postmodern ones, which can most clearly be seen when analyzing the connection between hedonists and lack of time. However, causes such as home experiences and lack of language skills also appear, which can characterize certain members of these groups, albeit to a lesser degree. When we turn to lifestyle groups with slow tempo, members and skippers—those losing ground and distressed—we cannot see even the possibility of travel, due to financial problems, more limited needs, and a rejection of travel ("I don't like traveling"). For segments in the middle, such as security seekers, the resigned, recognition seekers, and digital consumers, a duality of causes can be observed as closely connected. Recognition seekers and digital consumers tend to be characterized by anxiety, insecurity, and a fear of new situations and foreign places, as opposed to security seekers and the resigned, who reported lack of companionships and the inability to stay idle. Ill health seems to be independent of lifestyle group membership, as this cause exerts a negative effect on all groups.

## 5. Discussions

It is increasingly exciting to investigate the causes and motivations of non-purchase/ non-consumption, especially because this area is just touched on briefly in tourism-related research; however, understanding the latent or potential demand is an up-to-date and highly important part of the research of several sub-disciplines of tourism, such as demand analysis, carrying capacity studies, or the study of the effects of mass tourism. The results of the article provide a more in-depth analysis of the study of non-tourism, especially by realizing the opportunity of using the LifestyleInspiration model [70–72], which proved to be a useful tool in order to better understand who the non-travelers are and what the real causes of non-traveling are. From the point of view of consumer motivations, the international literature identified two basic cases: opting out and skipping [81]. Skipping is generally related to a state of lack (of time, of money, of knowledge, of companionship, etc.), whereas opting out is rooted in one's own informed decision. This latter case is called postmodern non-purchase/non-consumption, deriving from a notion related to a concept, a life skill, a religion, or conscience, which today appears in a marked manner (such as ignoring certain goods and ingredients) in eating habits [82] and food trends. Such phenomena can be explained by behavior triggered by social come-down [83,84] or generational distancing [85]. The extension of eating habit concepts is gaining ground for other categories of goods (vegan leather goods, such as car seats or violin strings, or even content, such as halal browsers), including tourism (vegan hotels, green hotels) [86].

As far as tourism is concerned, the phenomenon of non-purchase/non-consumption can be explained by a number of factors, especially in the skipping category. Our paper has shown that according to the results of the Hungarian representative sample, 43.3% of respondents traveled at least once for holiday or relaxation in the year prior, with the remaining 56.7% not engaging in any travel of a tourism nature. Our additional analyses on this non-traveling sub-sample showed that out of the classic skippers, 30.3% of non-travelers cited lack of money and 14.9% lack of time as reasons, thus helping us to reveal the main causes of classic skipping, coupled with lack of companionship (6.4%) and of language skills (6.2%).

Our analysis of differences related to generations, financial status, residence, and education revealed that, in line with initial expectations, there was a positive correlation between financial state and travel and education and travel. The higher one's income and qualifications, the more likely that one engages in tourism, and the less likely that we see cases of classic skipping. Nevertheless, evidence was provided that respondents, after being able to cover the most basic needs, pay for tourism even under more modest financial conditions. With residence location in the focus, it was proved that most non-traveling respondents reporting lack of time or money were from rural areas and the countryside, resulting from the time-consuming nature of the rural lifestyle as well as the more modest financial status of those living in the countryside. As far as generations are concerned, health issues,

the cumbersome nature of travel, and lack of comfort tended to more significantly affect senior citizens than they did the younger generations.

In addition to examining the causes of non-tourism and the characteristics of people absent from classical tourism, the opportunities offered by virtual tourism was also highlighted, finding that several megatrends (such as digitization and sustainability) support its development and influence [40]. It has also been posited, proving the literature feedback, that virtual tourism may be able to lessen the pressure of excessive tourism at the analyzed Hungarian multitude as well [41]. Moreover, we have found that although a mere 9.1% of respondents reported absolutely positive affinity (as opposed to 39.5% reporting absolute rejection), this nearly 10% ratio, in our view, may be reason enough for exploiting such opportunities. This theme is closely related to generational differences as well: virtual tourism has divided the respondents, with members of the younger generation being clearly more open to it.

The analysis of the relationship between lifestyle groups (identified by value orientation and life tempo dimensions) and non-tourism has been revealed, which is a novel approach in the tourism- and marketing-related research. Using correspondence analysis, the analysis proved that citing one's health status as cause of skipping is independent of respondent status. For groups with a speedier life tempo, we have identified significantly higher levels of travel, although for them we have also found evidence of affinity toward virtual tourism. All of this suggests that the two categories, in groups that are capable of making a choice, are complementary rather than mutually exclusive.

## 6. Theoretical and Practical Implications

The authors of the article consider the analysis of social groups averse to tourism an exciting theme both empirically and theoretically. It is our conviction that such studies can deepen the knowledge and research methodology for answering, on the one hand, the traditional question, critical for the tourism industry, "Can the number of people entering tourism be increased?" and on the other hand, another question focusing on the sustainable aspect, "Who are the non-travelers, and can they be considered as a form of preserving and safeguarding natural and built environment?". For this, the authors provided a method of how to identify classic as well as postmodern skipper groups in order to better and more deeply understand the exact reasons and the segments of society for non-tourism and non-travel.

As research implications, the authors believe that the methodology and results of the present study can be used for the actors of the tourism supply market and for decision makers as well, because if we can understand who the non-tourists or non-travelers are, we can then, on the one hand, traditionally determine the latent tourism potential of a tourism destination from the demand side, but we can also receive information on the specific market segment, which could contribute to sustainable tourism mostly because of the postmodern causes for non-traveling. In practice, this research further deepens the knowledge on understanding non-consumption, deriving basically from resignation, and can promote the creation of new market strategies.

## 7. Limitations of the Research

A certain limitation of the study is that it relies on one empirical piece of research from Hungary; however, the introduced methodology can be used in general for identifying and evaluating the non-travelers. Of course, the achieved results show the opinion and approach of the respondents at the exact time they were questioned (2018) and, since consumer preferences can change from time to time, the survey shall be repeated or updated periodically in order to receive up-to-date information on consumer preferences related to tourism. Another limitation can be connected to the methodology of the questionnaire survey, since the researchers had to rely on the answers of the respondents, and it was impossible to decide whether entirely true or possibly false data was provided.

## 8. Future Research Directions

As for the future research directions, the research shall be integrated with deep interviews with decision makers and decisive actors of the tourism industry as well, focusing on the further analysis of resignation, since it strongly supports the realization of sustainable objectives. The authors also plan to apply the introduced methodology and research design in different regions/countries as well and to update the survey periodically in order to see and understand the possible changes in the consumer preferences related to tourism. The demonstrated research design can also be used for other markets of the economy as well.

## 9. Conclusions

Our research highlights that, in order to better understand who the non-travelers are and what the real causes of non-traveling are, a lifestyle segmentation analysis can be a beneficial form of research design. Using correspondence analysis, apart from the well-known classic reasons, we have identified those groups where the causes of non-travel are mainly due to postmodern causes, such as virtual tourism. In this manner, these new findings reveal those segments where potential demand is present and can be harnessed by new approaches in tourism. Apart from that, those segments where real demand is not present can also be identified.

**Author Contributions:** Conceptualization: M.T., J.C., Á.N., and L.D.D.; methodology: Á.N. and M.T.; software: Á.N.; validation: Á.N. and M.T.; formal analysis: M.T., J.C., Á.N., and L.D.D.; investigation: M.T., J.C., and Á.N.; resources: L.D.D. and J.C.; original draft preparation: J.C. and L.D.D.; writing—review and editing: J.C. and L.D.D.; visualization: Á.N.; supervision: M.T. All authors have read and agreed to the published version of the manuscript

**Funding:** The research was financed and supported by the EFOP-3.6.1-16-2016-00004 "Comprehensive Development for Implementing Smart Specialization Strategies at the University of Pécs" and by the Higher Education Institutional Excellence Program of the Ministry of Human Resources, Hungary within the framework of the 4th topic of the University of Pécs, entitled "The enhancement of the role of the domestic enterprises in the reindustrialization of the nation" (20765-3/2018/FEKUTSTRAT).

**Conflicts of Interest:** The authors declare no conflict of interest. The funders had no role in the design of the study; in the collection, analyses, or interpretation of data; in the writing of the manuscript; or in the decision to publish the results.

## Appendix A. Sample Characteristics

| Gender | | | Generation | | | Region | | |
|---|---|---|---|---|---|---|---|---|
| male | 973 | 48.6% | younger (18–29 years) | 462 | 23.1% | Central Hungary | 239 | 11.9% |
| female | 1028 | 51.4% | middle-aged (30–59 years) | 1082 | 54.1% | Central Transdanubia | 216 | 10.8% |
| | | | older (>60) | 457 | 22.8% | Western Transdanubia | 203 | 10.1% |
| Financial Situation | | | Education | | | Southern Transdanubia | 183 | 9.2% |
| very good | 151 | 7.5% | elementary school | 271 | 13.5% | Northern Hungary | 231 | 11.6% |
| good | 790 | 39.5% | vocational training | 681 | 34.0% | Northern Great Plain | 299 | 15.0% |
| fair | 869 | 43.4% | vocational high school | 406 | 20.3% | Southern Great Plain | 256 | 12.8% |
| bad | 97 | 4.8% | secondary grammar school | 239 | 11.9% | Budapest | 372 | 18.6% |
| very bad | 22 | 1.1% | higher technical school | 125 | 6.2% | Type of Settlement (if not Budapest) | | |
| | | | BA/Bsc | 191 | 9.5% | county town | 423 | 21.1% |
| | | | MA/Msc | 78 | 3.9% | city or town | 654 | 32.7% |
| | | | not answering | 10 | 0.5% | village | 552 | 27.6% |

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
