# Peer review of "Can We Understand Non-Tourism as a Form of Sustainable Tourism? The Role of Lifestyle and Motivations behind Non-Traveling Based on the Hungarian Example"

_sustainability, doi:10.3390/su12187353_

Round 1

Reviewer 1 Report

Abstract:

  • The abstract is well written and the purpose of this study is clear. However, methodology, the limitation, and implication of this study needs to be incorporated in the abstract section.

Introduction:

The introduction section is through and well organized. However, I have presented some suggestion for improving the current study.

  • On page 2, line 47, please add the rationale for choosing Hungarian context for this study.
  • On page 2, line 46-47, please add citation for this claim.

Materials and Method:

  • The Materials and method section is very weak.
  • Any ethics clearance has been obtained before the field work? Please mention it.
  • The method section could be improved by having sub-titles (sample selection, procedures, etc.) that guide the readers through the entire research process in a more logical sense.
  • The author(s) need to justify why quantitative method is the ideal over qualitative method.
  • Consider issues inherent on the reliability and validity of the study to make it more robust. For example, incorporate data triangulation measures to strengthen data validity.
  • A sampling design is supported by presenting Hungarian location. The author(s) may want to:
  • Amplify more deeply the extent to why Hungarian location bear significance and justify the choice.
  • More clearly justify sample populations. Present the sample population and sample choice perhaps in table format.
  • Address the extent to which sample bias was dealt with.
  • Please add the survey questionnaire in Appendix.

Results:

  • Please provide your theoretical frameworks section before the materials and methods section separately.
  • Many citations in the text are more than 10 years of old. Please incorporate recent citation wherever possible.
  • On page 4, line 147 to 149 and line 152 to 154, require citation for the arguments. Please revise it.
  • On page 4, line 157 to 158, please add citation for your argument.
  • On page 4, line 173 to 181, can be shifted to the materials and method section.
  • On page 5, line 192, regarding K-means cluster analysis can be incorporated in the material and method section.
  • I think results could be presented separately in a clearer and consistent way. The author(s) need to do a better job for describing the whole results.
  • On page 5, line 271, about ‘Likert scale’ has not been discussed in the measures and method section. Please incorporate it.
  • On page 8, line 308. There should be one full stop instead of two full stop at the end of ‘question word’. Please revise it.
  • Outcome of the Figure 4. has not been discussed in the results section. Please incorporate it.
  • On page 9, line 333 to 351, author(s) are requested to place the real decimal numbers as per Table 1. For example, career builders should read 67.9% instead of 68. Please revise accordingly for all the areas.
  • On page 10, line 365 to 366, please add the word as ‘see Fig.5 after end of the sentence.

Discussion and conclusions:

I found the discussion sections of the paper also rather weak and without adequate depth.  The   following concerns are there:

  • Provide sufficient and in-depth discussion section. To improve these results for readability and sufficiency the author(s) may wish to support the results with recent references particularly from line 405 to 432 section.
  • Your conclusion section is very weak.
  • On page 11, line 443, author(s) discussed about ‘sustainability aspect’, however, this aspect has not been mentioned anywhere in the paper. Please discuss this aspect in the theoretical framework section.
  • There are no practical and theoretical contribution in this paper. You should have separate theoretical and practical implications section.
  • Author(s) also need to mention clearly the limitation of the current study.

Suggestion: It would be good to have separate discussion section and then implication (theoretical and practical) section, and limitation, and future research section.

Over all the language of this paper is very clear and the paper is written very well way. Addressing the above issues will definitely enhance your work. I am looking forward to reading your revised paper. Good luck to your work.

Author Response

Dear Honoured Reviewer! First of all, we would like to thank you very much for your review. With this support, together with the opinion of the other 3 reviewers, we believe that we could review our article, making it much better in context and in style. Trying to react all of your remarks:

  • The Abstract has been rewritten according to the comments: limitation and implication of the study has been incorporated.
  • The Introduction has been rewritten according to the comments, new citations were included, rationale has been added.
  • The Materials and methods has been rewritten according to the comments.
  • The Results chapter has been rewritten according to the comments: more than 30 relevant and new citations have been included deepening the theoretical background of the article (also at the theoretical background chapter).
  • Mistyping has been improved.
  • In the text the topic of sustainability has been strengthened both at the theoretical parts and at the Discussion.

The structure of the text has also been changed due to the opinion of another reviewer. Discussion and Conclusions were updated and reviewed accordingly and new chapters such as Research implications, limitations and future research have been added.

The Authors of this article are once again thankful for your valuable work and we hope that now the article has been improved to be able to accept it for publication.

Reviewer 2 Report

Dear Editor and Author(s),

Thank you for the opportunity to read the paper entitled: Can We Understand Non-Tourism as a Form of Sustainable Tourism? The Role of Lifestyle and Motivations Behind Non-Traveling Based on the Hungarian Example.

The topic of this paper is interesting, however, a lot of provided information needs more than just an edit. If the author(s) think that they will be able to fix all issues and rewrite the manuscript it might be reconsidered in the future as a new one. 

Main issues requiring amendments include the following:

  1. Structure of the manuscript:

1a. The text reads much like the author had an interesting idea and drawn together ideas and references without having a clear vision of how to structure and build up a paper. As a consequence, the manuscript reads a bit like a preliminary draft that the author is struggling with to put it into a paper that flows logically.

1b. Methodology section is missing. The reader does not know what author wants to achieve (and how?). Author admits in the Methods: Statistical methods have been applied …’ – what kind of statistical methods? Also, you have said that ‘The survey, employing a sample of 2,000 people …’ – how you have chosen them? In conclusion, no general recommendations would be possible if we do not know how you identify those. And probably this is a reason why the Discussion and the Conclusion section are missing. Overall, current discussion section is mainly repetition of results. 

1c. The text is rather a brief description of the current situation in Hungary, than identifying a problem and attempting to find a solution.

  1. Literature:

2a. The literature used in the Introduction in this manuscript is far from standards. Well, it's ok for introduction if you would have the background and describe there what we know and what we don’t know in the matter of your topic.

2b. Lack of references which should refer to the topic, i.e. non-tourism and sustainability see e.g.: McKercher: www.sciencedirect.com/science/article/abs/pii/026151779390046N

  1. Text formatting and language:

3a. The manuscript is very messy, please take a look on the Abstract:

‘In our view, the topic of non-tourism or non-travelers is especially interesting from the point of view of sustainable tourism and destination carrying capacity researches, so the study of postmodern causes besides classic ones holds unique potential in the research of sustainable tourism processes as well’.

This sentence is very difficult to understand. Also, you cannot use ‘in our view’. Our research showed … or so. ‘In our view’ is like opinion.

3b. You need to ask someone for English proofreading.

Please let me give you some advice to improve your future version of the manuscript. Please note that you don’t have to follow the structure placed below and you can change the section titles. All this is to help you to prepare proper paper with logical structure.

Abstract

There is a lack of some necessary information. The abstract should be in a few sentences also give something of applications. The abstract is not just what is an article, but also shows the achievements (results). The abstract must contain the aim of the paper, a sentence about methods.

Introduction

Your Introduction is a bit messy and does not contain all the necessary information. Literally speaking, the Introduction must answer the questions: What was I studying? Why was it an important question? What did we know about it before I did this study? And, How will this study advance our knowledge? Try to focus on a wider perspective and use your case study as an example. Introduction should have no more than one page, so the reader will be able to read it quickly.

Background

I would like to recommend you to develop a proper background. This part should begin with defining a topic to a wider audience. Thus, the background of your study will provide context to the information discussed throughout the research paper.

Furthermore, this section should discuss the theoretical aspects by involving the background of the theories published previously in the research literature and also focus on the ambiguities arose in these works.

  1. Some parts of it you have already in section: 3.1. The theoretical framework of non-tourism and non-travel. Furthermore, a framework do not belong to results as results should be based on previously prepared framework.

Methodology

The research methodology is the specific procedures or techniques used to identify, select, process, and analyze information about a topic. In a research paper, the methodology section allows the reader to critically evaluate a study's overall validity and reliability.

Results

The results section of the research paper is where you try to "sell" the findings of your study based upon the information gathered as a result of the methodology you applied. The results section should simply state the findings, without bias or interpretation, and arranged in a logical sequence. The Interpretation and your opinion you can place in the next section - the Discussion.

Discussion

The discussion part is missing in this paper. That is why I recommend you follow my suggestions to develop proper Introduction and Background. This will allow you to conduct proper Discussion. The purpose of the discussion is to interpret and describe the significance of your findings in light of what was already known about the research problem being investigated, and to explain any new understanding or fresh insights about the problem after you've taken the findings into consideration. The proper discussion should be connected to the introduction. Thus proper Introduction and Background are necessary.

Conclusion

A few sentences describing your main findings. Please note that most of the readers will read Introduction and Conclusion first before they will decide if your paper is worth reading in a full.

Author Response

Dear Honoured Reviewer! First of all, we would like to thank you very much for your review. With this support, together with the opinion of the other 3 reviewers, we believe that we could review our article, making it much better in context and in style. Trying to react all of your remarks:

  • First of all the structure of the text has been changed based on your suggestions.
  • The Abstract has been rewritten according to the comments: limitation and implication of the study has been incorporated.
  • The Introduction has been rewritten according to the comments, new citations were included, rationale has been added.
  • The Literature chapter has been extensively improved: new aspects and more than 30 new citations have been included deepening the theoretical background of the article. The topic of sustainability was also included here.
  • In connection with the language, we went through the text again and tried to correct the hardly understandable contexts.
  • The Materials and methods has been rewritten according to the comments.
  • The Results chapter has been rewritten according to the comments: the mentioned citations have been updated.
  • Mistyping has been improved.
  • In the text the topic of sustainability has been strengthened both at the theoretical parts and at the Discussion.
  • Discussion and Conclusions were updated and reviewed accordingly and new chapters such as Research implications, limitations and future research have been added.

The Authors of this article are once again thankful for your valuable work and we hope that now the article has been improved to be able to accept it for publication.

Reviewer 3 Report

My comments are given in the attached file.

Author Response

Dear Honoured Reviewer! First of all, we would like to thank you very much for your review. With this support, together with the opinion of the other 3 reviewers, we believe that we could review our article, making it much better in context and in style. Trying to react all of your remarks:

  • First of all the structure of the text has been changed based on your suggestions.
  • The Abstract has been rewritten according to the comments: limitation and implication of the study has been incorporated.
  • The Introduction has been rewritten according to the comments, new citations were included, rationale has been added.
  • The Literature chapter has been extensively improved: new aspects and citations have been included deepening the theoretical background of the article. The topic of sustainability was also included here.
  • In connection with the language, we went through the text again and tried to correct the hardly understandable contexts.
  • The Materials and methods has been rewritten according to the comments.
  • The Results chapter has been rewritten according to the comments: new citations have been included deepening the theoretical background of the article (also at the theoretical background chapter).
  • Mistyping has been improved.
  • In the text the topic of sustainability has been strengthened both at the theoretical parts and at the Discussion.
  • Discussion and Conclusions were updated and reviewed accordingly and new chapters such as Research implications, limitations and future research have been added.
  • We need to add, that the newly suggested topics were not included in the text because of the word limits and also since we think that they do not need to be included in the context. We hope you understand it.

The Authors of this article are once again thankful for your valuable work and we hope that now the article has been improved to be able to accept it for publication.

Reviewer 4 Report

The paper is generally well-written and the chosen topic is relevant given the scope and themes of the journal. I am relatively happy with the analysis but there are some major weaknesses, as follows:

  1. Abstract - this reads very informally and is better for an Introduction. The Abstract should focus more explicitly on the topic, the purpose of the paper, methods employed and findings.
  2. Introduction - relatively fine but there is a need to put aim and objectives of the paper in a more explicit way. I would also recommmend to provide a justification of the chosen case study
  3. Literature Review - it is just not there as such. The topic is quite interesting and can be linked with some other areas.
  4. Methods - ok but needs more details here. Why did you choose this method? How does it fit with the research aim and objectives?

Overall, the paper has a potential but needs more work. English is not a major issues here but a proof-read by a native speaker would enhance the quality of the paper for sure.

Author Response

(The authors gave the same response as above.)

Round 2

Reviewer 1 Report

  • The authors has been requested to add the definition of “LifestyleInspiration model”. However, it has not been addressed.
  • K-mean cluster analysis should be included in the methodology section and that has not been addressed.
  • Likert scale perspective has not been addressed in the methodology section.
  • The results from page number 8 in line 327, page 9 in line 331, and page 9 in line 337 needs to be presented in a separate table and that has not been addressed.
  • Survey questionnaire has not been added in the Appendix section for this research.
  • Why stratified sampling has been used? Needs justification for this method.
  • Any ethics clearance has been obtained before the field work? This aspect has not been addressed
  • The authors need to justify why quantitative method is the ideal over qualitative method. Need to be addressed
  • Consider issues inherent on the reliability and validity of the study to make it more robust. For example, incorporate data triangulation measures to strengthen data validity. Has not been addressed.
  • More details explanations are required for ‘excluding sampling bias and how’?
  • The paper’s title indicated about Sustainable tourism. However, this aspect has not been indicated in this paper. A greater section of sustainable tourism literature has been missed. This aspect needs to be added in the theoretical section.
  • Paper needs professional editing.
  • Addressing the above issues will definitely enhance your work.
  • Good luck to your work.

Reviewer 2 Report

Dear Editor and Author(s),

Thank you for the opportunity to read the revised paper entitled: Can we understand non-tourism as a form of sustainable tourism? The role of lifestyle and motivations behind non-traveling based on the Hungarian example.

I must say that you have done a good job and the manuscript is far much better than it was previously.

As I have mentioned the topic of this paper is interesting.

Main issues requiring amendments include the following (no of the line according to MDPI system):

Title. I do not like the word example – is correct but please consider:

Can we understand non-tourism as a form of sustainable tourism? A Hungarian case study on the role of lifestyle and motivations behind non-traveling

(14) A certain limitation is that our research is based on the case of Hungary …

This is not a limitation – you are using a case study to show / explain some part of reality. Please change that.

Some sentences need to be rewritten – please note that the English language does not like long sentences. Please check your manuscript and change split sentences like the following ones into two or three smaller ones:

(289) Correspondence analysis is a useful tool to uncover the relationships among categorical variables, it is a multivariate graphical technique designed to explore relationships – and this is exactly what we have aimed at in our research, to explore the relationship between lifestyle groups and causes of non-travel.

Eg:

Correspondence analysis is a useful tool to uncover the relationships among categorical variables. It is a multivariate graphical technique designed to explore the relationship between lifestyle groups and causes of non-travel. This is exactly what we have aimed at in our research.

Same comment:

As research implications, the authors believe that the methodology and results of the present study can be used for the actors of the tourism supply market and for decision-makers as well, because if we can understand who are the non-tourists or non-travelers, we can then on the one hand traditionally determine for a tourism destination the latent tourism potential from the demand side, but we can also receive information on the specific market segment which could contribute to sustainable tourism mostly because of the postmodern causes for non-traveling.

It’s a 90 words sentence!

Furthermore, in some sentences, you are using ordinary language:

(438) It is increasingly exciting to investigate the causes and motivations of non-purchase / non-consumption especially because this area is just touched on briefly in tourism-related researches, however understanding the latent or potential demand is an up-to-date and highly important part for the research of several sub-disciplines of tourism, such as demand analysis, carrying capacity studies or the study of the effects of mass tourism.

Eg. ‘It is increasingly exciting’ ?

Investigating the causes and motivations of non-purchase and non-consumption is an important part of the tourism economy research. This topic was only briefly touched by tourism-related researches. However understanding the latent or potential demand is an up-to-date and highly important part for the research of several sub-disciplines of tourism, such as demand analysis, carrying capacity studies or the study of the effects of mass tourism.

(262) This part is not necessary – please delete it:

(262) After providing an overview of the relevant literature and advancing the theoretical framework of the theme, the paper investigates the relationship of non-tourism, non-travel and the previously introduced lifestyle groups. In order to identify these groups and the aimed relationship, a quantitative analyses is needed, which can be later on further assessed through qualitative methods.

(267) 3.1. Sample selection

Now your description n of sample selection looks correct. I would like to suggest you add Appendix A to the text as a Table. The information provided on it is also interesting.

(327, 331 at al) Please check the p-value. You have that p<0.000 ?

(494) The authors of the article consider the analysis of social groups averse to tourism an exciting theme both empirically and theoretically.

(just the authors) The authors consider ….

(519) Another limitation can be connected to methodology of the questionnaire survey, since the researchers have to rely on the answers of the respondents, and it is impossible to decide whether entirely true or maybe false data was provided.

This is not a limitation! You should use the so-called 'the test question', that is, ask the same question in a different way in a different part of the questionnaire. In that way, you will be able to exclude dishonest respondents.

At this stage you can not do that so erase this sentence from your paper.

Conclusion: You should focus on your main findings.

Reviewer 3 Report

My suggestions have not been fully implemented. However, the article looks much better.

Reviewer 4 Report

The manuscript has been substantially revised and I am now happy with the changes made.